# Optimization of Alkali Extraction and Properties of Polysaccharides from *Ziziphus jujuba cv.* Residue

**DOI:** 10.3390/molecules24122221

**Published:** 2019-06-14

**Authors:** Zongxing He, Yulian Zhu, Xingyu Bao, Liuxin Zhang, Nan Li, Gonglingxia Jiang, Qiang Peng

**Affiliations:** 1College of Food Science and Engineering, Northwest A & F University, Yangling 712100, China; hezongxing1125@163.com; 2College of Innovative and Experiment, Northwest A & F University, Yangling 712100, China; pinellia1999@163.com (Y.Z.); baoxingyu98@163.com (X.B.); 13991875686@163.com (L.Z.); 3Key Laboratory of Environment of Change and Resources Use in Beibu Gulf, Ministry of Education (Nanning Normal University), 175 Mingxiu East Road, Nanning, Guangxi 530001, China; nli@gxtc.edu.cn; 4Beijing Engineering and Technology Research Center of Food Additives, Beijing Technology & Business University (BTBU), Beijing 100048, China

**Keywords:** *Ziziphus jujuba cv.* Muzao residue, lye extraction, polysaccharides, extraction optimization, physicochemical property

## Abstract

*Ziziphus jujuba cv*. Muzao is a plant widely cultivated in the Yellow River Basin of China. It has nutritional and healthcare functions, in which polysaccharides are the main components of its bio-functions. In order to make effective use of *Ziziphus jujuba cv.* Muzao residue resources and explore new functional food ingredients, the polysaccharide (ZJRP) from *Ziziphus jujuba cv.* Muzao residues were extracted by sodium hydroxide, and the optimal extraction conditions of ZJRP were obtained by the response surface method. The basic composition and antioxidant effects of ZJRP were determined. The results showed that ZJRP has significant antioxidant activity, mainly reflected in the high DPPH radical scavenging rate, which may be related to their high content of galacturonic acid and the extraction method. In addition, the rheological and thermal properties of ZJRP were respectively determined by a rheometer and differential scanning calorimetry (DSC), indicating that they have shear thinning properties and good thermal stability. Results showed that the alkaline extraction method can be used as a potential technique for extracting ZJRP with high antioxidant activity, and ZJRP can be further explored as a functional food ingredient.

## 1. Introduction

*Ziziphus jujuba cv.* Muzao belongs to the genus *Ziziphus* (family Rhamnaceae). It is mainly cultivated in the Yellow River Valley areas of China (Yulin, Shaanxi Province and Lüliang, Shanxi Province) [1], and is considered to be the most important *Ziziphus* species that is nutritious and widely planted [2,3]. Modern pharmacological studies have shown that *Ziziphus jujuba cv.* Muzao has the functions of anti-oxidation, anti-inflammation, sedation, liver protection, immunity, and liver protection [4,5]. *Ziziphus jujuba cv.* Muzao contains various chemicals including vitamin C, polysaccharides, phenolic acids (hydroxycinnamic acid and benzoic acid), flavonoids (flavonols and flavan-3-ols), triterpenic acid, and nucleosides [2], and rich polysaccharides are the main active ingredients [3]. Plant polysaccharides have a variety of pharmacological effects, and most of them are nontoxic and natural. They are ideal choices for health foods and medicines. In recent years, increasing research attention has been paid to plant polysaccharides [6,7]. So far, there have been reports on the water-extracted *Ziziphus jujuba cv.* Muzao polysaccharides, which indicate that the *Ziziphus jujuba cv.* Muzao polysaccharides obtained by water extraction have immunoregulatory, anti-oxidation, anti-tumor, and hypolipidemic activities, etc., and have gradually been applied in food processing [4,5,8]. However, we thought that after water extraction, some of the *Ziziphus jujuba cv.* Muzao residue may still have some active substances that are not used, and up to now, there is no report on the alkali extraction of polysaccharides from *Ziziphus jujuba cv.* Muzao residue [3,8]. Polysaccharides extracted under different pH conditions are different in many respects, such as chemical composition, biological activity, etc. [9]. By changing the pH, the polysaccharide components which are not easily soluble in water are dissolved, which is beneficial for the reuse of industrially produced scraps.

Therefore, the purposes of this study are to: (i) Optimize the extraction of polysaccharides from *Ziziphus jujuba cv.* Muzao by sodium hydroxide solution, and promote the reuse of *Ziziphus jujuba cv.* Muzao residue; and (ii) determine the basic composition and function of *Ziziphus jujuba cv.* Muzao residue polysaccharides (ZJRP) to explore them as new functional food ingredients.

## 2. Results and Discussion

### 2.1. Single-Factor Experimental Evaluation

The influences of liquid-to-material ratio (5:1–25:1), sodium hydroxide concentration (0.05–0.15 M), temperature (50–90 °C), and extraction time (0.5–2.5 h) on the extraction yield of ZJRP were investigated by the classical ‘‘one factor at a time” methodology (Figure 1A–C). In this method, all factors remain unchanged during the experiment except those under study. This type of test revealed the effects of the selected variables under the given conditions. The extraction process was subsequently evaluated at different liquid-to-material ratios (5:1–25:1). Other factors were fixed, with temperature at 30 °C, sodium hydroxide concentration at 0.1 M, and 0.5 h was taken as the extraction time (Figure 1A). The figure demonstrated the extraction efficiency of polysaccharides increased with the increase of the ratio of liquid to material. The single-factor results show that when the ratio of liquid to material was 20:1, the extraction effect of polysaccharides was the best. Thereafter, with the liquid-to-material ratio was further increased, the extraction rate of polysaccharides gradually decreased. This result is similar to the reported result [10]. The effect of extraction temperature on ZJRP yield was investigated by different extraction temperatures (50–90 °C). As shown in Figure 1B, the extraction efficiency increases with the increase of extraction temperature in the range of 50–80 °C, and the maximum extraction rate is reached at a temperature of 80 °C. After this temperature, the ZJRP extraction yield decreased. These findings demonstrate that elevated temperatures promote the release of polysaccharides. However, when the temperature reaches the sensitive level of polysaccharide extraction, these macromolecules may degrade [11]. Zou et al. also demonstrated that prolonged processing at elevated temperatures may result in degradation of biomacromolecules such as polysaccharides, colorants, and polyphenols [12]. The extraction efficiencies of polysaccharides at different concentrations were determined by using the concentration of sodium hydroxide (0.05–0.15 M) as the experimental factor. The experimental results are shown in Figure 1C. It can be concluded from the figure that the extraction rate of ZJRP increases with the increase of the concentration of the alkali solution when the concentration of the alkali solution is low (0.05–0.1 M), and reaches the maximum value when the concentration of the alkali solution is 0.1 M, and then decreases with the further increase of the concentration of the alkali solution. The reason for this trend may be that with the increase of alkali concentration, more polysaccharides can be released by reacting with sodium hydroxide. However, when the concentration of the lye reaches a certain value, it will react with the polysaccharides. The polysaccharides are degraded, so the extraction effect is reduced. For the time factor, the change of the extraction effect in the range of 0.5–2.5 h was determined. The results were compared using the Duncan method by SPSS 20.0, and there was no significant difference in the extraction effect after the extraction time reached 0.5 h. We considered that 0.5 h was short enough, so we directly selected 0.5 h for follow-up research. The extraction effect did not change significantly after the time reached 0.5 h, indicating that the sodium hydroxide solution can quickly extract the residual polysaccharides in the *Ziziphus jujuba cv.* Muzao residue.

### 2.2. Optimization of Extracting Conditions by the Box–Behnken Design

According to the single-factor experimental evaluation above, three variables and three levels were selected to optimize the extraction conditions of ZJRP. On the basis of BBD (Box–Benhnken design), 17 studies were conducted to study the extraction conditions of ZJRP. The experimental parameters and ZJRP yields (%) for 17 runs are shown in Table 1. The predicted response Y of the ZJRP yield can be derived from the following equation:Y = 2.55 + 0.059X_1_ − 0.061X_2_ + 0.078X_3_ − 0.064X_1_X_2_ + 0.038X_1_X_3_ − 0.055X_2_X_3_ − 0.24X_1_^2^ − 0.22X_2_^2^ − 0.028X_3_^2^
where Y is the predicted yield of ZJRP (replaced by absorbance), X_1_ is the liquid-to-material ratio encoding parameter, X_2_ is the lye concentration encoding parameter, and X_3_ is the extraction temperature encoding parameter.

The results were then subjected to the F-test and ANOVA analysis by BBD to estimate the significance and applicability of the regression model (Table 2). The results showed that the regression model of ZJRP yield was highly significant (*P* < 0.01). The model’s coefficient of determination (R^2^) was 0.988, indicating that the experimental data were in good agreement with the predicted results. In addition, the adjusted adj R^2^ value (0.972) represents a high degree of correlation between the observed and predicted values. The coefficient of variation (C.V.), which indicates the standard deviation as a percentage of the average and determines the reproducibility of the model, was 1.450%, which is less than 5.00% [13]. The significance of the coefficients can be checked by the P value: The smaller the P value is, the more significant the corresponding coefficients are [14]. As shown in Table 2, linear coefficients (X_1_, X_2_ and X_3_), cross-product coefficients (X_1_X_2_), and quadratic coefficients (X_1_^2^ and X_2_^2^) were found to be highly significant (*P* < 0.01).

### 2.3. Response Surface Analysis

The influence of a single variable and the interaction between two variables can be seen in the two-dimensional (2D) contour plot (Figure 1D–F). These figures show the statistical significance of the interaction between the two test variables. A circular contour plot indicates that the interaction between the two variables is not significant, while an ellipse contour line or saddle contour line reveals that the interaction between the two variables is significant [15].

Based on the Design Expert software, the optimal condition for extracting ZJRP is as follows: The liquid-to-material ratio was 21.2:1, the lye concentration was 0.09 M, the extraction temperature was 90 °C, and the maximum predicted extraction yield was 2.629 (replaced by absorbance). Considering the convenience in the actual operation, the actual conditions were changed slightly: The liquid-to-material ratio was 20:1, the lye concentration was 0.09 M, and the extraction temperature was 90 °C. In the previous single-factor test, the extraction effect of the polysaccharides was reduced after the temperature reached 80 °C, and the reason why the temperature increase and the yield did not decrease in the response surface analysis result may be due to the interaction between these factors. Subsequently, the optimization result was verified, and the polysaccharide solution was obtained using the optimum extraction conditions. The absorbance value of the solution was determined to be 2.632 (repeated three times, using SPSS 20.0 for analysis of variance, *P* < 0.01), which was in agreement with the optimization result. The yield was calculated to be 5.1%, based on the mass ratio of the obtained polysaccharides (after depigmentation treatment and freeze-drying) to the original *Ziziphus jujuba cv.* Muzao powder. The polysaccharides obtained from *Ziziphus jujuba cv.* Muzao with water extraction had a total yield of 3.82% relative to the dried jujube fruit power, which was less than the yield of ZJRP production.

### 2.4. Physical and Chemical Properties of ZJRP

#### 2.4.1. Chemical Analysis

The phenol–sulfuric acid assay showed the carbohydrate content, and the protein was measured by the Bradford method. The total sugar and protein contents in ZJRP (obtained through optimized conditions) were 63.7% and 9.0%, respectively. The content of uronic acid was 31.5%. The total sugar content and uronic acid content of ZJRP are lower than water-extracted crude polysaccharides (67.29% and 49.55%, respectively) from *Ziziphus jujuba cv.* Muzao [8]. The reason may be due to the difference in extraction methods, and other non-sugar ingredients may be extracted under alkaline conditions. The polysaccharides extracted by the base may contain more ashes and are more susceptible to the Maillard reaction [16]. ZJRP is composed of rhamnose, arabinose, glucose, galactose, glucuronic acid, and galacturonic acid, in mass percentages of 3.3:3.0:2.6:1.9:1.0:25.1, respectively. Other studies have also found that some monosaccharides are common in *Ziziphus jujuba cv.* Muzao fruit and in the polysaccharides extracted from the same variety of Z. fruit, including arabinose, glucose, galactose, rhamnose, and galacturonic acid [8]. The sugar content would vary, probably due to differences in extraction methods and purification steps [8].

#### 2.4.2. FT-IR Analysis

The configuration of the polysaccharides can be elucidated by FT-IR spectroscopy. The spectrum of ZJRP is shown in Figure 2A. The band at around 3350 cm^−1^ was due to OH stretching. The signal near 2943 cm^−1^ corresponded to the stretching of -CH and -CH_2_ groups, and the two bands at 1609 and 1412 cm^−1^ corresponded to the asymmetric and symmetric vibrations of the -COO (carboxylate) structure, respectively [17,18]. The reason may be the presence of uronic acid. The band near 1096 cm^−1^ represents the β-glycosidic bond between sugar units. The signal close to 1240 cm^−1^ was caused by the S=O stretching vibration, which indicated that it may contain the sulfate group. The absorption at 895 cm^−1^ was related to the C–H variable angular vibration of the β-anomeric carbon [19]. Finally, the absorption band at around 638 cm^−1^ indicated that C–CO–C stretching vibration may exist in ZJRP [8]. The characterization of ZJRP with FT-IR identified absorption peaks typical for polysaccharides.

#### 2.4.3. Thermal and Rheological Properties

The ZJRP heat flux as a function of temperature is shown in Figure 2B. As can be seen from the figure, the exothermic peak and endothermic peak of ZJRP are at 123.56 and 256.22 °C, respectively, which indicates that ZJRP melts at 123.56 °C. The onset of weight loss at 256.22 °C (representing the onset of oxidation or decomposition) indicates that the polysaccharide has good thermal stability. The thermal behavior and transition temperature of the sample are caused by differences in the material structure and functional groups. The physical and chemical changes of polysaccharides during heat treatment produce unique curves for a given polysaccharide [20]. The heat of fusion is 11.0 J/g for the extracted polysaccharide, which may be related to the molecular weight (Mw) of the polysaccharide, the degree of methylation, and the galacturonic acid content [21]. Typically, the endothermic peak and heat of fusion reflect the ability to retain water, which is related to the hydrophilic groups of ZJRP. The reason for this may be due to the action of sodium hydroxide. In addition, the curve has a smaller endothermic peak, which may be caused by a higher protein content.

Regarding the rheological properties of ZJRP, as shown in Figure 3A, the extracted polysaccharide showed pseudoplastic fluid characteristics similar to those of other polysaccharides formed by their random coil formation [22]. With the increase of shear rates, the viscosity of the polysaccharide decreases rapidly, which could be related to the molecular weight of the polysaccharide, the amount of galacturonic acid, and the degree of methylation [23,24]. The viscosity characteristics of the extracted *Ziziphus jujuba cv.* Muzao residue polysaccharide indicated that ZJRP aqueous solution (2%, *w*/*w*) has shear-thinning characteristics as a food thickener. If the extracted pectin polysaccharides are added to a paste or pulp food, these foods may flow more easily when shaking or applying other external force [23].

In order to determine the linear viscoelastic region of ZJRP, strain scanning was performed with the deformation values of 0.01%−100%, as shown in Figure 3B. It can be seen that the loss modulus (G’’) of the polysaccharide was always higher than the storage modulus (G’), and within a certain strain range, G’ and G’’ remained basically unchanged with the increase of strain, which is the linear viscoelastic region of ZJRP. When the strain exceeded the range of the linear viscoelastic region (especially after more than 10%), both G’ and G’’ of ZJRP decreased. Then, 1% strain was selected to determine the frequency dependences of G’ and G’’. As shown in Figure 3C, the loss modulus of ZJRP increases rapidly with the increase of frequency, while the storage modulus of ZJRP increases first and then decreases, but is always low. This shows that the viscosity of ZJRP is dominant in the viscoelasticity, which is also consistent with previous results, and ZJRP has shear fluidity, with the potential to be added as a thickener to paste or pulp foods [23,25]. Different from many other pectin polysaccharides, the elastic properties may be due to the low degree of polymerization, loose molecular structure, less intermolecular chain interaction, and low viscoelasticity of *Ziziphus jujuba cv.* Muzao residue polymers extracted by alkali solution [26,27].

### 2.5. In Vitro Antioxidant Activity of ZJRP

There are many different in vitro assays for determining the antioxidant activity of natural products, such as hydroxyl radical scavenging capacity determination, DPPH free radical scavenging capacity determination, iron reduction capability determination, etc. [28].

Hydroxyl radicals produced by the human body may play an important role in tissue damage in inflammatory sites of oxidative stress-induced diseases [29]. DPPH is a stable free radical with the ability to become an antimagnetic molecule by accepting electrons or hydrogen ions [30]. Therefore, the DPPH free radical scavenging experiment and the OH free radical scavenging experiment were used to evaluate the antioxidant activity of ZJRP in this study. As shown in Figure 4A,B, the clearance rates of both free radicals by the polysaccharide and ascorbic acid increased with increasing dose. Ascorbic acid has a good scavenging effect on OH and DPPH at a concentration of 1 mg/mL. ZJRP had no significant effect on OH scavenging activity in the experimental range, but the scavenging effect increased with the increase of dose. It is noteworthy that ZJRP has a DPPH free radical scavenging rate of 70.67% at a lower concentration (1 mg/mL), and is still rising. Although the antioxidant effect of ZJRP is not outstanding compared with ascorbic acid, it can be judged that ZJRP has good antioxidant activity.

Figure 4C shows the chelating abilities of polysaccharides to ferrous ions. ZJRP effectively isolates iron ions in a dose-dependent manner. When the polysaccharide content reached 5 mg/mL, the chelation effect reached 53.8%, while the EDTA standard achieves a 99.9% sequestration effect at 100 μg/mL [31]. The presence of protein in the polysaccharide component may reduce the chelating ability for ferrous ions, so the higher protein content in ZJRP may be an important reason for its chelation ability [31]. In a previous study, it has been pointed out that the presence of hydroxyl or carboxyl groups in compounds is the main reason for the chelating abilities of metal ions [32]. The galacturonic acid content of the polysaccharide and the presence of the alcoholic hydroxyl group in the polysaccharide component may contribute to the chelation of the metal ion [33]. Analysis of significance was done using SPSS 20.0.

The activities of polysaccharides are inseparable from their complex molecular structure, so the reason for the antioxidant activity of ZJRP remains to be further studied in the future.

## 3. Materials and Methods

### 3.1. Materials and Reagents

*Ziziphus jujuba cv.* Muzao (dried fruit with seed) was provided by a loess plateau experimental orchard of Jia County, Shaanxi, China, in 2018. The obtained *Ziziphus jujuba cv.* Muzao fruits were carefully washed, halved, and oven-dried (model DHG-9203A, JingHong, China) at 60 °C for 24 h. The dried samples were crushed and sifted in a 60-mesh screen to obtain powder samples. The samples were stored in a glass dryer with a silicone desiccant at room temperature for further analysis. Standard monosaccharides (*l*-rhamnose, *d*-arabinose, *l*-fucose, *d*-xylose, *d*-mannose, *d*-glucose, and *d*-galactose) were purchased from Sigma Chemical Co. (St. Louis, MO, USA). All the chemicals and solvents were of analytical grade, such as sodium hydroxide and ascorbic acid. Deionized water was generated with a Milli-Q Water Purification System (Millipore, Bedford, MA, USA).

### 3.2. Preparation of Ziziphus jujuba cv Residue

The dried *Ziziphus jujuba cv.* Muzao powder was extracted by deionized hot water (liquid-to-material ratio = 10:1) at 60 °C for one hour (twice), and then the non-polysaccharide components detected by the phenol–sulfuric acid method were centrifuged to remove the supernatant, and the *Ziziphus jujuba cv.* Muzao residue was obtained. This process was to simulate the acquisition of the industrially produced scraps, such as jujube residues remaining after the production of jujube beverages.

### 3.3. Extraction of Polysaccharides

The effects of different extraction times on the extraction rate of dried *Ziziphus jujuba cv.* Muzao residue under different temperatures (50–90 °C), sodium hydroxide concentrations (0.05–0.15 M), and liquid-to-material ratios (5:1–25:1) were studied. The value of the extraction rate was determined by using the phenol–sulfuric acid method, where 1 mL of 5% phenol solution was added to 1 mL of the polysaccharide solution, and then 5 mL of sulfuric acid (98%) was added, reacted for 30 min, and then the absorbance was measured (model UV7 spectrophotometer, METTLER TOLEDO, Zurich Switzerland) at 490 nm [34]. On the basis of single-factor experiments, the temperature, liquid-to-material ratio, and alkali concentration were determined to be independent variables under the condition of an extraction time of 30 min, depending on whether the influence on the extraction rate was significant or not. All experiments were repeated three times and the data were processed by SPSS 20.0. After the extraction, the polysaccharide solution was concentrated and lyophilized to prepare a 5% aqueous solution of polysaccharide, and this was added to 10% hydrogen peroxide at 50 °C for 2 h for decolorization, and then concentrated and lyophilized for 24 h to obtain ZJRP.

### 3.4. Box–Behnken Design

The software Design Expert (Version 8.0.6, Stat-Ease, Minneapolis, MN, USA) was employed for experimental design, data analysis, and model building [35]. A Box–Behnken design was designed with three variables: Liquid-to-material ratio (X_1_), alkali concentration (X_2_), and extraction temperature (X_3_), each of which had three levels. Then, we defined a quadratic polynomial model to accommodate the response (% polysaccharides yield):$$Y=\beta_{0}+\sum_{i=0}^{n}\beta_{i}X_{i}+\sum_{i=0}^{n}\beta_{ii}X_{ii}+\sum_{i\neq j=0}^{n}\beta_{ij}X_{ij}$$
where Y represents the measured response variables and β_0_ is a constant which represents the intercept coefficient. β*_i_*, β_*i*2_, and β*_ij_* are respectively the linear term, the quadratic term, and the interaction term [36].

### 3.5. Chemical Analysis

The total sugar content of ZJRP (obtained through optimized conditions) was determined by the phenol–sulfuric acid method, with glucan (molecular weight of 7 W) as the standard [34]. The protein content was measured by the method of Bradford, using bovine serum albumin as the standard [37]. With glucuronic acid as the standard, the content of uronic acid was evaluated by the Blumenkrantz method [38].

### 3.6. Monosaccharide Analysis

Determination of monosaccharide composition in ZJRP was done according to the method reported by Wang et al. with slight modification. [39]. The monosaccharide composition was released from ZJRP by acid hydrolysis with 2M TFA at 121 °C for 2 h. The hydrolyzate was evaporated (at 60 °C) using a rotary evaporator, and then distilled water was added and evaporated to dry and remove trifluoroacetic acid [8]. After acid hydrolysis, Na_2_CO_3_ was added, and the mixture was kept in a water bath at 30 °C for 45 min, and then newly prepared 4% NaBH_4_ solution was added, and the mixture was kept at room temperature for 1.5 h. Then, 25% acetic acid was added to neutralize excess NaBH_4_ until no bubbles were produced. In order to remove sodium lactone ions, which significantly affect the reduction of uric acid, the mixture was loaded onto a cationic exchange resin column and eluted with water, and then evaporated dry under reduced pressure using a rotary evaporator (60 °C, model RE-201D 2L RE-52A, China). Methanol and evaporate (60 °C) were added under reduced pressure to remove boric acid ions, which affect the formation of aldehydes and alcohols. Thereafter, the dried mixture was heated at 85 °C for 3 h to completely convert the aldonic acid to the aldonic acid lactone. Next, the sample was ready for the next derivatization reaction with n-propylamine and acetic anhydride. One mL pyridine and 1 mL n-propylamine were added, and the mixture was sealed and heated at 55 °C for 30 min, and then cooled by a rotary evaporator to 55 °C for drying. Then, 1 mL pyridine and 1 mL acetic anhydride were added and kept at 95 °C for 1 h, and then cooled and dried. The mixture was dissolved using dichloromethane, and then the sample was filtered (0.45 μm pore size) before the gas chromatography (GC) analysis.

The monosaccharide derivatives were detected with GC (Shimadzu 2014C; Shimadzu Corp., Kyoto, Japan) coupled to a high-performance capillary column DB-17 (30 mL × 0.25 mm ID, 0.25 μm film thickness; Agilent Technologies, Santa Clara, CA, USA). The flow rate of carrier gas (N_2_) was 1.5 mL/min with an injection volume of 1 μL. The injector and detector temperatures were respectively 250 and 280 °C. The temperature program was as follows: 180 °C (2 min), heating at 6 °C/min until the temperature reached 210 °C, temperature maintained at 210 °C (2 min), then heating at 0.3 °C/min until the temperature reached 215 °C, temperature maintained at 215 °C (20 min), and then heating at 8 °C/min until the temperature reached 240 °C, and temperature maintained at 240 °C (10 min) [8,39,40]. The standard monosaccharide samples were treated by the same derivatization method to detect tested samples.

### 3.7. Analysis of FT-IR Spectra

The IR spectra of ZJRP were recorded by the KBr disk method [8]. The polysaccharides were crushed with KBr powder, and then pressed into pellets for FT-IR (Vertex 70, Bruker, Germany) measurement in the frequency range of 4000−400 cm^−1^ with a resolution of 4 cm^−1^ to detect functional groups.

### 3.8. Thermal Analysis

The thermal properties of ZJRP were determined by differential scanning calorimetry (DSC Q2000, TA Instruments, New Castle, DE, USA) [41]. Four mg of polysaccharide powder was loaded into an aluminum crucible and referenced to an empty aluminum crucible. In the N_2_ environment with a gas flow rate of 50 mL/min, the heating rate was 10 °C/min, and the temperature scanning range was 30−300 °C. The phase transition temperature and phase transition enthalpy were measured.

### 3.9. Rheological Analysis

The *Ziziphus jujuba cv.* Muzao polysaccharide solution for rheological tests was prepared by mixing the sample with distilled water (2%, *w*/*w*). The prepared solution was measured by a rheometer (AR1000, TA instruments, USA) with a 40 mm parallel plate. The sample solution was enslaved to stable shearing at 25 °C, and the shear rate ranged from 0.02 to 100 s^−1^ [42]. The storage modulus (G’) and loss modulus (G”) of the sample solution were determined using an oscillation measurement. A strain sweep (0.01−100% strain at 1 Hz) was applied to test the linear viscoelastic region of the sample. The frequency dependences of G’ and G” were determined by a frequency sweep (0.1–10 Hz at 1% strain).

### 3.10. In Vitro Antioxidant Activity Analysis

#### 3.10.1. DPPH Radical Scavenging Capacity

The in vitro antioxidant activity of ZJRP was determined by measuring DPPH free radical scavenging activity by the previously reported method [8], with some modifications. One mL of different concentrations of polysaccharide solution (1−5 mg/mL) was mixed with 1 mL DPPH–methanol solution (0.1 mM), and protected from light for 30 min [30]. Ascorbic acid (1 mg/mL) was used as a positive control (as shown in Figure 4).

#### 3.10.2. Hydroxyl Radical Scavenging Activity

The hydroxyl radical scavenging activity of ZJRP was measured as previously reported [23]. Two mL of 9 mM ferrous sulfate (FeSO_4_) and 2 mL of 9 mM salicylic acid–ethanol solution were mixed with 2 mL of ZJRP (1–5 mg/mL). The reaction was started by adding 2 mL of 8.8 mM H_2_O_2_ solution, and the absorbance (model UV7 spectrophotometer, METTLER TOLEDO, Zurich Switzerland) was measured at 510 nm after standing at 37 °C (water bath heating) for 1 h. Ascorbic acid (1 mg/mL) was used as a positive control (as shown in Figure 4).

#### 3.10.3. Chelating Ability of Ferrous Ion

According to the reported method, the chelating ability of Fe^2+^ was slightly modified [31]. One mL of ZJRP (1–5 mg/mL) was added to 0.1 mL of 2 mM ferrous chloride (FeCl_2_) and 3.7 mL of distilled water. After mixing for 30 s, 0.2 mL of 5 mM ferrozine solution was added and reacted at 25 °C for 10 min. The absorbance value was measured at 562 nm. Ascorbic acid (1 mg/mL) was used as a positive control (Shown in Figure 4).

## 4. Conclusions

In this study, ZJRP was extracted from *Ziziphus jujuba cv.* Muzao residues by response surface methodology (RSM) with sodium hydroxide solution for the first time. The optimal extraction conditions were as follows: The liquid-to-material ratio was 21.2:1, the lye concentration was 0.09 M, the extraction temperature was 90 °C, and the extraction time was 0.5 h. Using the improved method, the extraction rate reached 5.09%. Chemical analysis showed that the total sugar content and protein content of ZJRP were 63.7% and 9.0%, respectively, and the uronic acid content was 31.5%. Analysis of the monosaccharide composition by gas chromatography showed that ZJRP were composed of rhamnose, arabinose, glucose, galactose, glucuronic acid, and galacturonic acid, in mass percentages of 3.3:3.0:2.6:1.9:1.0:25.1, respectively. DSC analysis illustrated ZJRP has good thermal stability. The viscosity characteristics of the extracted *Ziziphus jujuba cv.* Muzao residue polysaccharides indicated that ZJRP aqueous solution (2%, *w*/*w*) has shear-thinning characteristics as a food thickener. The antioxidant activity assay showed that the OH and DPPH free radical scavenging activities of ZJRP were 22.0% and 82.5%, respectively, at 5 mg/mL. At the same concentration, the chelation ability of ZJRP to ferrous ions reached 53.9%. This study provides important information for a new and efficient method for extracting polysaccharides from industrially produced scraps, which has broad application prospects and can be widely used in the industry in the future.

## Figures and Tables

**Figure 1 molecules-24-02221-f001:**
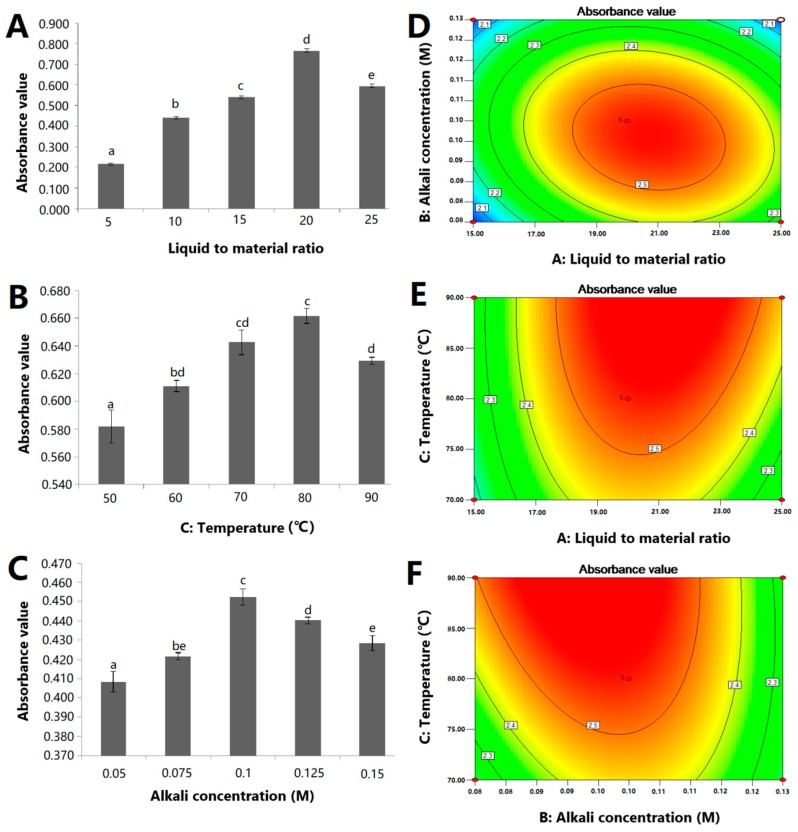
Effects of singlefactor and two-factor interactions on the polysaccharide extraction yield of *Ziziphus jujuba cv.* Muzao residue polysaccharides (ZJRP). (**A**) Liquid-to-material ratio, (**B**) extraction temperature, (**C**) alkali concentration, (**D**) alkali concentration and liquid-to-material ratio, (**E**) extraction temperature and liquid-to-material ratio, and (**F**) extraction temperature and alkali concentration.

**Figure 2 molecules-24-02221-f002:**
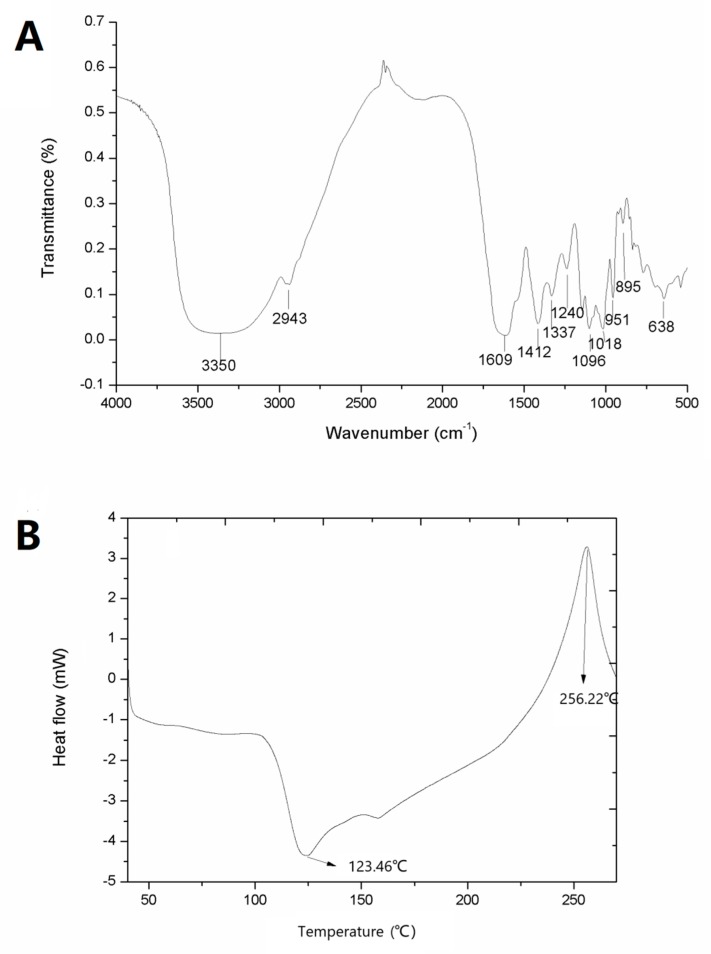
FT-IR spectra and differential scanning calorimetry (DSC) thermogram of ZJRP (obtained through optimized conditions). (**A**) FT-IR spectra, (**B**) DSC thermogram.

**Figure 3 molecules-24-02221-f003:**
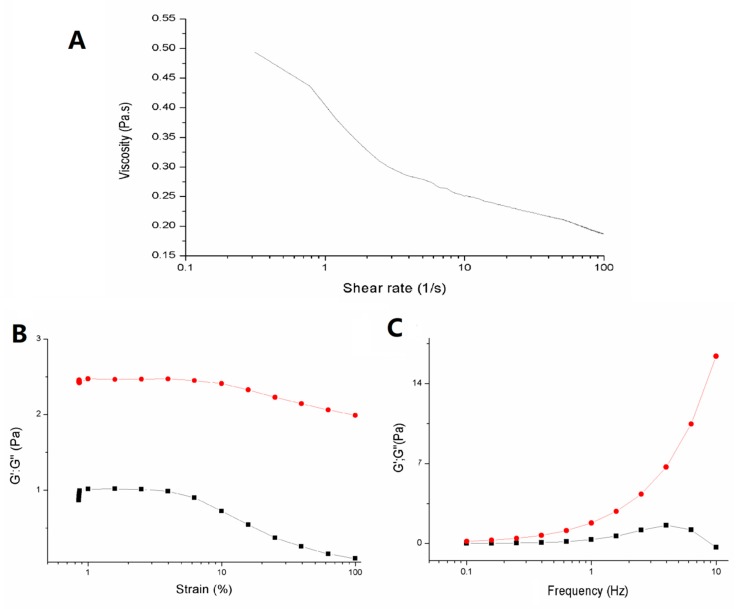
Rheological analysis of ZJRP (obtained through optimized conditions). (**A**) The flow behavior of ZJRP. (**B**,**C**) Storage modulus (G’) and loss modulus (G’’) of ZJRP (**B**, strain sweeps at 0.1 S^−1^ frequency of modulus. **C**, frequency sweeps at 1% strain.). The black squares represent the storage modulus (G’) and the red dots mean the loss modulus (G’’).

**Figure 4 molecules-24-02221-f004:**
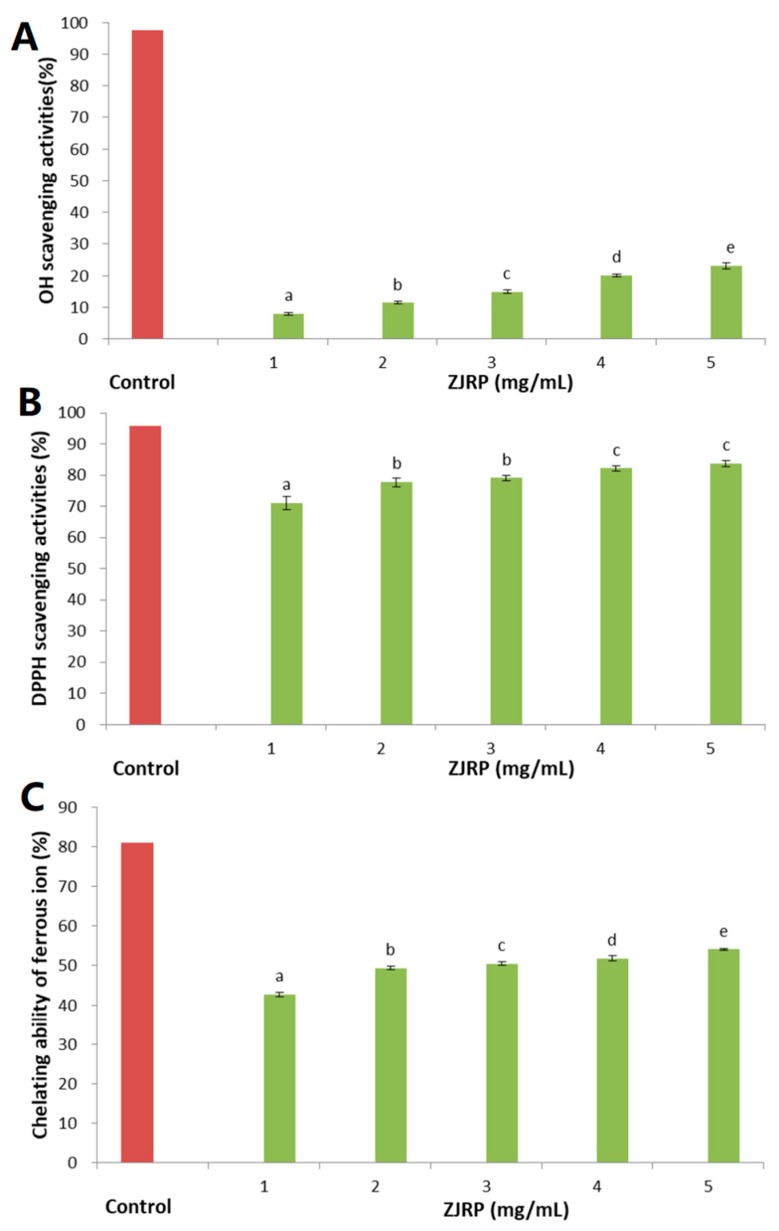
In vitro antioxidant capacity of ZJRP (1–5 mg/mL). (**A**) OH radical scavenging activity, (**B**) DPPH radical scavenging activity, and (**C**) chelating ability of ferrous ion. Ascorbic acid (1 mg/mL) was used as a positive control. (The different letters in the figures represent significant differences from other results, *P* < 0.05).

**Table 1 molecules-24-02221-t001:** Box–Behnken central composite design for independent variables and their responses.

Run	Independent Variables			Actual Value	Predicted Value
	Liquid to Material Ratio	Alkali Concentration (M)	Extraction Temperature (°C)		
1	15.00	0.08	80.00	2.01	2.02
2	20.00	0.10	80.00	2.57	2.55
3	20.00	0.08	90.00	2.48	2.50
4	20.00	0.08	70.00	2.22	2.23
5	15.00	0.10	70.00	2.20	2.18
6	20.00	0.13	90.00	2.28	2.26
7	25.00	0.10	90.00	2.43	2.45
8	20.00	0.13	70.00	2.23	2.22
9	20.00	0.10	80.00	2.53	2.55
10	20.00	0.10	80.00	2.55	2.55
11	15.00	0.13	80.00	1.99	2.03
12	25.00	0.08	80.00	2.31	2.27
13	20.00	0.10	80.00	2.54	2.55
14	25.00	0.10	70.00	2.19	2.22
15	25.00	0.13	80.00	2.03	2.02
16	20.00	0.10	80.00	2.56	2.55
17	15.00	0.10	90.00	2.29	2.26

**Table 2 molecules-24-02221-t002:** F-test and ANOVA analysis of the response surface quadratic model.

Source	Sum of Squares	df	Mean of Square	F Value	*p*-Value
Model	0.630	9	0.070	62.590	<0.001 **
X_1_	0.028	1	0.028	24.870	0.002 **
X_2_	0.030	1	0.030	26.370	0.001 **
X_3_	0.049	1	0.049	43.290	<0.001 **
X_1_X_2_	0.016	1	0.016	14.460	0.007 **
X_1_X_3_	0.006	1	0.006	5.140	0.058
X_2_X_3_	0.012	1	0.012	10.760	0.013 *
X_12_	0.250	1	0.250	223.830	<0.001 **
X_22_	0.200	1	0.200	181.220	<0.001 **
X_32_	0.003	1	0.003	2.980	0.128 *
Residual	0.008	7	0.001		
Lack of Fit	0.007	3	0.002	11.180	0.021
Pure Error	0.001	4	0.000		
Cor Total	0.640	16			
R^2^	0.988				
Adj R^2^	0.972				
C.V%	1.450				

* *P* values < 0.05 were considered to be significant. ** *P* values < 0.01 were considered to be highly significant.

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
