# Peer review of "Optimization of Alkali Extraction and Properties of Polysaccharides from Ziziphus jujuba cv. Residue"

_molecules, 2019, doi:10.3390/molecules24122221_

Round 1
Reviewer 1 Report
The experimental design is adequately presented, although it can be improved in some parts. For example lines 123-127 are not clear.
The novelty should resid in the NaoH extraction of polysaccharide, according to the authors. Some comments about previous extractions with other "solvents" should be commented. A comparison with polysaccharides extracted in other conditions would be of benefit for the readers.
Line 58: the ratio 20:1 was kept for the subsequent investigations?
Lines 78-80 are not clear.
The manuscript lacks comments. Results are presented but they are poorly discussed.This aspect should be improved.
Author Response
Thank you very much for your timely and helpful suggestions! We have already revised the manuscript from your comments. You can see these changes in the modified manuscript and Response to Reviewer 1 Comments.

Reviewer 2 Report
The paper is interesting but it needs major revision.
Title: polysaccharide or polysaccharides? its properties - suggest sodium hydroxide, I suggest to change for: Alkali extraction/Extraction optimization and properties of polysaccharides from Ziziphus jujuba cv residues – or similar to avoid misunderstandings
Firstly, the name of Ziziphus jujuba should be carefully unify in all text (jujube or jujube?, Italic style or not, bold or normal fonts, capital or small letters, short or full name, etc. l.13, 17, 34, 35, 41, 42, 46, 50, 137, 139, 222, 228, 241, table 1 headings, )
Secondly, there are a lot of punctuation errors, spaces are in excess or are missing, distance between numbers and units is often missing (l.3 l. 46, figure 1 D,E,F axis titles, fig. 2B, fig 4 A,B,C, l.118, l.143, 158, 190, 230, l.231, 233, 247, 248, 263, 273, 274, 276, 281, 292, 295, 296, 300, 312)
mL should be corrected (l. 300, figure 4 A,B,C)
Thirdly, similar tense should be used (Present or Past) to describe aim of the study (l.46-47), methods (l.226 were of analytical grade) and results (eg. l.61 was investigated, l.63 is reached; l.71 was degraded…is reduced, l79 there is no …l.80 was no longer..; l.148 indicates, l.150 indicated; is determined l.235, were determined l.237; sugar… is l.250, protein ... was l.251; should be were respectively l.276, should be was instead of were l.313) Past Tense should be applied to description of the results because tests have been done already.
For all figures – font size and style should be unify as required by Journal
l. 16 - polysaccharide or polysaccharides?
l.27 Abbreviations: the only ZJRP is explained, but there are also other abbreviations in the text – should be explained (eg. BBD l.88, CRP l.102, EDTA l.210, TFA l.258)
l.43 – often different – in what sense? Pls. specify
l.50, 69 – sodium hydroxide as reagent as well as concentration is not mentioned in Materials and methods section
figure 1
l.90 – dot is missing
l.93, 247 where – with small letter
table 2 – it should be ta same number of decimal places in one table
l.110 what is b? pls. correct
l.118 – 21.2, but in Conclusion 21
l.127 – after depigmentation and freeze-drying – but there are any information about these methods in Materials and methods section
l.131, 132 – 63.7%, 31.5% - but in Conclusions 63.71 and 31.46
l.137 – other studies – but there are no references cited
l.141, 279, 280 – FTIR should be unify (FT-IR?)
l.153 should be thermogram because only one is presented, title should be concise, now it is doubled; it should be specified in the title for what sample curves are presented
l.180 – Figure 3C with capital letter
l.190 s-1 – pls. correct
l.203 ZJRP had no significant effect – but in fig 4 there are different letters indicating significant differences, I think (pls. add description in the title what means different letters)
l.206 – is Vc ascorbic acid – if yes pls. add in the figure 4 title, moreover ascorbic acid wasn’t mentioned in Materials and methods section and in Results
l.210 EDTA standard - not mentioned in Materials and methods section
figure 4 – x-axis Concentration of what? Pls. specify. Why separate analysis was set for ZJRP and Vc? If Vitamin C was used it should be clearly described in graph and in Results section
l.223 - should be oven-dried, model and producer’s data should be added
l.225 – samples were stored in a dryer? If true, in what conditions? Or correct
l.230 – pls. change as in l.224
l.231, 233 – dregs or residue – pls. unify
l.234 – what alkali and what concentration? – this information is crucial for the paper, including title, because from methods and description of the results it is not clear what solvent was used and what were conditions of extraction
l.235 – method should be described in brief (especially conditions)
l.236-237 – range of each factor or should be described (similar as in Results section), time of 30 minutes is mentioned but in l.78 time range was 0.5-2.5 hours – pls. specify
l.242 – these two sentences should be combined as one
l.255 – style of this sentence should be improved
l.258 rotary evaporator – model and producer’s data should be added
l.259 – from the part 1 – but it is not specified what is part 1, - pls. be more concise
l.265 – what methanol – not mentioned before
l.268 should be heated instead of sealed, what amounts of reagent were added? Pls. specify
l.272-274 – style should be improved
l.276 – should be - temperature program was as follows:..
l.278 – should be tested instead of mixed
l.280 – polysaccharides were crushed? Was the sample in a solid state? Any prior description
l.281 - model and producer’s data should be added, any references?
l.302, 308, 314 – should be specified, Vc is applied on figures
l.308 – equipment should be specified, model and producer’s data should be added
l.311 chelating effect of what?
l.316 should be 4.4.
l.323 ZJMP?
l.326 – pls. specify that 2% solution was used; has better shear … - better than what? Pls. specify
l.330 – from industrially produced scraps – but Authors used fresh whole fruits, this conclusion is not following the tests results presented
l.348 – should be full journal name
l.405, 430, 432, 434, 445 – pls. correct titles for small letters
l.452 – description of sample availability is missing
Author Response
Thank you very much for your timely and helpful suggestions! We have already revised the manuscript from your comments. You can see these changes in the modified manuscript and Response to Reviewer 2 Comments.

Round 2
Reviewer 1 Report
The following is as it was in the original version, some mistakes regarding verbs.:
Subsequently, the optimization result was verified, and 138 the polysaccharides solution was obtained using the optimum extraction conditions were measured 139 to have an absorbance value of 2.632 (repeated three times, using SPSS 20.0 for analysis of variance, 140 P<0.01), which was in agreement with the optimization result.
Still in section 3.10.2 and 3.10.3 some verbs nnot corrected
Author Response
Thank you very much for your suggestions. We have revised and responded to your suggestions point-by-point, you can see in the PDF.
Thank you again for reviewing our manuscript during your busy schedule.

Reviewer 2 Report
The paper has been corrected significantly but I have some more remarks before publication:
l.17, 265 Sodium should be with small letters
l.19, l.23 should be: results showed
l.43, 44, 46 Ziziphus Jujuba should be in Italic style
l. 55 to explore as a new…
l.72 pls. remove brackets
l.87 should be: were degraded… effect was reduced
l.110 should be Table 2
l.112 should be: was 0.988, data were in good…
l.116 which was 1.450%
l.132 best extraction was found at 80°C (l.73) , why 90°C was selected as optimum – it is not clear
l.138 should be4: …solution….was measured
l.142 the polysaccharides were obtained
from l.145 as well as from l.286 – it should be clearly stated that presented results are for optimum? sample or pls give extraction conditions applied in figure 2 and 3 captions
l.153 should be: and are more…
l.155 should be: other studies
l.157 should be: sugar content would vary
l.161, 175 – pls unify the tense: was shown or is shown
l.168, 169 repetition: stretching vibration
l.170 should be: typical for polysaccharides
l.172 should be: thermogram
l.183 related to these hydrophilic… which one? Pls specify
l.186 smaller… higher .. than what? Pls. rephrase
l.190 which is related – it should be: could be related if not confirmed by results but references
l.194 should be: are added
l.196 should be: was performed
l.199 1% strain was selected but results in Fig 3B starting form 20%, what was the reason?
l.222 at lower concentration – pls. specify
l.233 should be: in previous study, only one reference is presented
l.236 Analysis of significance was done using…
fig 4C – is it really true that b,c,d letters indicated significant differences (visibly these three results not differ)
l.250 should be sifted or separated instead of wrapped
l.292 was done is missing in this sentence
l.293 should be: reported by Wang et al.
l.301 should be: it was added
l.341, 347, 353 – pls. unify description of solution concentration
l.342, 349, 355 – should be: Ascorbic acid was used…
l.346 should be: 2 mL…..solution were taken
l.347 – should be: reaction was started
Author Response

(The authors gave the same response as above.)
